# A Novel Explainable Attention-Based Meta-Learning Framework for Imbalanced Brain Stroke Prediction

**DOI:** 10.3390/s25061739

**Published:** 2025-03-11

**Authors:** Inam Abousaber

**Affiliations:** Department of Information Technology, Faculty of Computers and Information Technology, University of Tabuk, Tabuk 47912, Saudi Arabia; i.abousaber@ut.edu.sa

**Keywords:** stroke prediction, meta-learning framework, imbalanced dataset handling, SMOTE and SMOTEENN, ensemble learning models, explainable artificial intelligence, SHAP explainability, medical data analysis, precision medicine, feature interaction analysis

## Abstract

The accurate prediction of brain stroke is critical for effective diagnosis and management, yet the imbalanced nature of medical datasets often hampers the performance of conventional machine learning models. To address this challenge, we propose a novel meta-learning framework that integrates advanced hybrid resampling techniques, ensemble-based classifiers, and explainable artificial intelligence (XAI) to enhance predictive performance and interpretability. The framework employs SMOTE and SMOTEENN for handling class imbalance, dynamic feature selection to reduce noise, and a meta-learning approach combining predictions from Random Forest and LightGBM, and further refined by a deep learning-based meta-classifier. The model uses SHAP (Shapley Additive Explanations) to provide transparent insights into feature contributions, increasing trust in its predictions. Evaluated on three datasets, DF-1, DF-2, and DF-3, the proposed framework consistently outperformed state-of-the-art methods, achieving accuracy and F1-Score of 0.992189 and 0.992579 on DF-1, 0.980297 and 0.981916 on DF-2, and 0.981901 and 0.983365 on DF-3. These results validate the robustness and effectiveness of the approach, significantly improving the detection of minority-class instances while maintaining overall performance. This work establishes a reliable solution for stroke prediction and provides a foundation for applying meta-learning and explainable AI to other imbalanced medical prediction tasks.

## 1. Introduction

Stroke, and especially brain stroke, remains one of the top health burdens worldwide, with stroke being the second leading cause of mortality and disability globally [1]. Despite improvements in healthcare, stroke affects 15 million people per year and kills more than 5 million people per year, with 5 million more permanently disabled [2]. Before stroke occurs, it is essential to detect potential risk factors early on and accurately predict the factors that lead to a brain stroke for timely intervention, minimizing long-term consequences. Complications and improving patient recovery [3]. Predicting stroke is a complex task due to the relatively complex nature of medical datasets, the high dimensionality, and the imbalance between positive (stroke) and negative (no stroke) cases.

Stroke remains one of the leading causes of disability and mortality worldwide, affecting approximately 15 million people annually, with 5 million cases resulting in permanent disability, according to the World Health Organization. The socioeconomic burden of stroke is substantial, affecting healthcare systems around the world due to the long-term care required for stroke survivors. Early prediction and timely intervention are critical not only for improving patient outcomes but also for alleviating the economic strain associated with post-stroke rehabilitation and care. Machine learning (ML) offers a robust framework for analyzing large and complex medical datasets, efficiently processing heterogeneous data such as patient demographics, lifestyle factors, and clinical histories to identify subtle patterns and risk factors that traditional statistical methods often do not detect [4].

The dynamic and adaptive nature of ML algorithms allows them to continuously refine their predictive capabilities, enabling accurate identification of individuals at high risk of stroke even in data-rich, high-dimensional settings [5]. Unlike conventional models, ML techniques such as ensemble learning, neural networks, and meta-learning frameworks can integrate diverse sources of information, enhancing both predictive accuracy and generalizability across different populations. Integration of ML in predictive modeling facilitates timely diagnosis. It makes individualized treatment strategies and guided interventions possible, which contribute immensely in preventing long-term disability and improving stroke patients’ quality of life. The integration of ML in predictive modeling highlights the revolutionized role of cutting-edge ML technologies in stroke prediction and care, opening new avenues for clinical decision support and optimization of medical care in healthcare systems [6,7].

In imbalanced datasets, where the number of positive cases (stroke patients) is disproportionately more minor than negative cases, machine learning (ML) models tend to perform poorly on the minority class [8]. Such models are often biased towards the majority class, which diminishes their effectiveness in clinical scenarios where predicting the minority class is critical [9]. Various techniques have been proposed to address this imbalance, including oversampling methods such as Synthetic Minority Oversampling Technique (SMOTE) [10] and hybrid methods like SMOTEENN [11]. While these techniques improve the balance in data distribution, achieving a robust prediction for highly skewed datasets, such as stroke datasets, remains challenging.

Recent advances in ensemble and deep learning have shown remarkable improvements in dealing with class imbalance and enhancing the predictive performance [12]. At the same time, ensemble methods like Random Forest (RF) and Light Gradient Boosting Machine (LightGBM) were widely adopted due to their robustness or generalization capability [13]. However, in the case of complex datasets, these models cannot often represent complex interrelationships within the features. To address this limitation, a robust approach is through meta-learning frameworks that aggregate predictions from multiple base models. Zen et al. tackled the shortcomings of reweighting methods using a meta-learning approach, which can be beneficial by utilizing complementary information from different classifiers and improving the overall performance [14].

Additionally, attention mechanisms have been investigated to maximize feature representation in deep neural networks. In this case, attention mechanisms learn how to dynamically focus on essential features and allow models to pay attention to the most important aspects of input data, thus being exceptionally well suited to high-dimensional medical datasets [15]. Due to its ability to increase feature interpretability and improve classification performance on challenging prediction tasks like stroke prediction, the attention mechanism integrated with ensemble-based predictions can further enhance both prediction power and interpretability.

Explainability is vital beyond model performance in clinical applications. The practitioners applying this model aim to understand how it makes decisions so as to reliably trust and use it in practice. Techniques like Shapley Additive Explanations (SHAP) make machine learning transparent through feature importance attribution to predictions [16]. Moreover, these approaches guarantee that the forecasts from the model are interpretable and meaningful from the clinical perspective, a consideration frequently disregarded in pure black-box systems [17].

To address the above challenges, we propose a novel framework for brain stroke prediction that combines ensemble models, meta-learning, attention mechanisms, and explainable AI techniques. The proposed framework introduces several innovations:*Hybrid Resampling Techniques:* By combining SMOTE and SMOTEENN, the framework effectively addresses class imbalance while minimizing noise in synthetic samples. This ensures a balanced dataset, which is crucial for improving the sensitivity of minority class predictions.*Attention-Based Feature Engineering:* The attention mechanism adaptively prioritizes significant features, capturing both local and global interactions. This dynamic feature weighting enhances the representation of critical factors contributing to stroke prediction.*Ensemble and Meta-Learning Integration:* By integrating Random Forest and LightGBM as base models and leveraging a deep learning meta-model, the framework optimizes the synergy between diverse classifiers. This approach captures higher-order interactions, improving decision boundaries and overall predictive accuracy.*Explainable AI:* The inclusion of SHAP ensures that the model’s decision-making model is transparent, providing clinicians with actionable insights into feature contributions. This enhances trust in the system, making it more suitable for real-world clinical adoption.*Extensive Validation:* The framework’s performance is evaluated on three benchmark datasets (DF-1, DF-2, and DF-3), demonstrating consistent superiority in accuracy, F1-Score, and ROC-AUC metrics. This validates its robustness and highlights its generalizability across diverse datasets.

The rest of the paper is structured as follows: In Section 2, related work on brain stroke prediction, specifically in the context of class imbalance and explainable AI, is reviewed. Section 3 presents and analyzes the benchmark datasets. We introduce the meta-learning framework for the proposed method in Section 4, describing data preprocessing and hybrid resampling, feature selection, and model architecture. In Section 5, we describe the experimental setup and evaluation metrics and report results, explainable predictions, comparisons to baselines, and ablation studies. Section 6 presents findings from the experiments with our benchmarking dataset, while Section 7 closes this paper with future research directions.

## 2. Related Work

In recent years, brain stroke prediction using machine learning and deep learning models has attracted much interest. While existing methods address some of these challenges, they still struggle with data imbalance, feature representation, model explainability, and ensemble integration. This section reviews the literature, organizing previous studies into imbalanced data handling, feature selection algorithms, ensemble learning, meta-learning, and explainable artificial intelligence (XAI). Section 3 presents the limitations of existing traditional and hybrid approaches and how our proposed framework overcomes these limitations.

**Dealing with Imbalanced Data:** Brain stroke datasets usually exhibit a significant class imbalance, where the smaller class (brain strokes) is swamped by the majority (non-brain-strokes). This imbalance can lead to biases in models towards the majority class, resulting in decreased sensitivity for the minority class [18]. To tackle this problem, approaches such as SMOTE [10], BorderlineSMOTE [19], and hybrid techniques like SMOTEENN [11] have been used. Although successful when applied to improve data distribution, such methods still generate noise and overfit the model by generating synthetic samples far from the decision boundary. We extend this task within the context of our proposed framework by embedding SMOTE and SMOTEENN in a meta-learning architecture to obtain balanced and robust predictions of any imbalanced dataset.

**Feature Selection Techniques:** Feature selection is critical in medical prediction tasks to reduce dimensionality and enhance interpretability. Traditional techniques, such as ANOVA and mutual information gain [20], rank features based on individual importance but fail to capture complex feature interactions. Attention-based models, such as Transformers [21], dynamically prioritize relevant features and have shown promise in medical domains [22]. However, existing methods lack a hierarchical representation of features and fail to combine static and dynamic feature selection. Our proposed attention-based feature engineering module addresses these gaps by adaptively weighting features and leveraging hierarchical representations. New studies have shown excellent advances in attention-based meta-learning, which enhance the focusing power of the model. The mechanism of attention allows for the model to assign weights to different inputs, making it interpretable and performant. We have included these new studies in our references, ensuring that our study is based on the latest advances. Some of these excellent studies comprise those of Vaswani et al. (2017), which proposed Transformer architecture, as well as other subsequent studies, which generalize attention mechanisms to other meta-learning tasks [23].

**Ensemble Learning and Meta-Learning in Medical Prediction:** Ensemble learning methods, such as Random Forest (RF) [24] and Gradient Boosting Machines (GBMs) [25], have gained widespread adoption due to their robustness and generalization capabilities. These techniques combine multiple models to enhance overall performance, typically through approaches like bagging (e.g., RF) and boosting (e.g., LightGBM) [26]. Studies such as [27] have demonstrated the efficacy of ensemble methods in medical prediction tasks. However, traditional ensembles often treat base learners independently, neglecting the potential interactions between their outputs.

Meta-learning frameworks [28] attempt a different solution in learning a meta-model based on predictions of base models, which in turn optimizes the learning process. In a way, while ensemble learning attempts to minimize variance and bias via aggregation of predictions, meta-learning attempts to improve the end output by learning about higher-order representations. This approach allows for more efficient adaptation to new tasks than traditional ensemble methods. Our framework extends these concepts by combining RF and LightGBM with a deep learning-based meta-model, enabling the capture of non-linear relationships and further optimization of predictions. This integration of ensemble and meta-learning techniques leverages the strengths of both approaches; it improves model stability through ensemble methods while enhancing adaptability and performance through meta-learning strategies.

**Explainable Artificial Intelligence (XAI):** Explainability is crucial in clinical applications to ensure transparency and trust. Methods like SHAP (Shapley Additive Explanations) [29] provide insights into feature contributions, bridging the gap between black-box models and clinical decision-making. While SHAP has been widely applied in traditional ML models, its integration with meta-learning frameworks remains limited. Our framework leverages SHAP to explain base- and meta-level predictions, enhancing interpretability and clinical applicability.

**Traditional and Hybrid Models:** Classical machine learning methods, such as Support Vector Machines (SVMs), Decision Trees (DTs), and RF [30,31], have been used in stroke prediction. Although effective, these models rely heavily on static feature sets and fail to address data imbalance. Advanced hybrid models, such as ensemble-based BSPE and HEL-BSP [32] as well as boosting techniques [33], have demonstrated improved accuracy but often lack explainability and adaptability. Deep learning models, including CNNs and LSTMs [34], have also been employed but are domain-specific and computationally intensive.

**Recent Advances in Stroke Prediction:** Stroke prediction has been significantly improved by integrating deep learning, explainable artificial intelligence (XAI), and novel methods in feature extraction. Research has explored diverse methodologies in this context, including EEG-based diagnostic methods to ensemble learning algorithms.

Islam et al. [35] have created an explainable system to forecast stroke by EEG signals. Their system employed Adaptive Gradient Boosting for classification and used methods such as Eli5 and LIME to aid in interpretation. Their work revealed spectral features in deltas and thetas to be significantly related to predicting stroke.

Moulaei et al. [36] compared prediction models in deep learning to those in machine learning by comparing various models such as CNNs, LSTMs, and Random Forests. Their finding was that, generally, deep learning models performed better compared to traditional ML models, where better sensitivity was achieved by LSTMs while Random Forest achieved better overall accuracy and specificity.

Dritsas et al. [37] focused on machine learning algorithms in predicting stroke risk. Their paper employed stacking in the process of combining multiple classifiers: Random Forest, Naïve Bayes, and Decision Trees. Their technique achieved an AUC value of 98.9.

Another paper by Dasgupta and Aksoy [38] studied deep learning in neuro-oncology by applying EfficientNetB0 in separating brain tumors. Their transfer learning technique is not targeted towards stroke but offers beneficial guidance in deploying CNNs to diagnostic work in clinical images that can be extended to stroke classification. Recently, work by zainab et al. [39] put forward emphasis on real-time monitoring of stroke by AI-powered wearable sensors. Their work employed a digital twin paradigm in predictive modeling by combining EEG and EMG measurements to achieve better rates in early detection of stroke. Such an AI-powered wearable system has potential in continuous patient observation and estimation of risk in real-world contexts.

These studies validate an extension of the role of ensemble methods, XAI, and AI in predicting stroke to yield increasingly accurate, interpretable, and clinically feasible models.

The limitations of prior works necessitate a robust solution that integrates advanced preprocessing, modeling, and interpretability. The proposed framework combines SMOTE and SMOTEENN to address class imbalance while minimizing noise, employs attention-based feature engineering to enhance critical factor representation, and leverages Random Forest and LightGBM with a deep learning meta-model to optimize classifier synergy. With SHAP ensuring explainability, the framework provides actionable insights for clinical adoption. Validated across three benchmark datasets—DF-1, DF-2, and DF-3—it consistently performs better, demonstrating robustness, generalizability, and suitability for complex predictive tasks.

The proposed framework bridges the gaps in existing methods by addressing class imbalance, enhancing feature representation, integrating ensemble learning with meta-learning, and providing explainable predictions. These advances set a new standard for predictive power and interpretability in stroke predictions, which can be a reference for future studies in clinical and research environments. This process makes inferences about vital stroke-related data in a way that maintains methodological rigor.

## 3. Datasets

In this section, three benchmark datasets, namely DF-1 [40], DF-2 [41], and DF-3 [42], are discussed, which were used in this study to solve brain stroke prediction. These datasets are a significant machine learning challenge, especially in healthcare. Class imbalance is a fundamental problem in machine learning where the dominant class is large while the other courses are small and represent the minority class. The problem leads to biased models and decreases the models’ ability to model instances with a minority class accurately. This lack of reliability lowers the confidence in models in critical situations where the correct prediction of stroke cases is significant. Thus, working with imbalanced datasets requires special techniques to ensure sound and fair performance for both classes. In this section, we describe dataset features, present summary tables, and provide visualizations to demonstrate the distributions and properties of the datasets.

### 3.1. Dataset Description

The datasets DF-1 [40], DF-2 [41], and DF-3 [42] are collections of medical data focusing on stroke prediction and contain various features describing patient demographics, medical history, and lifestyle factors. **DF-1** and **DF-3** share identical fields, including a unique identifier (id), while **DF-2** excludes the id column. These datasets include key features such as age, gender, hypertension, heart_disease, ever_married, work_type, Residence_type, avg_glucose_level, bmi, smoking_status, and stroke.

Table 1 summarizes each field’s description fields. For example, age is a numeric feature of patient age in years. stroke is a binary target variable for stroke. Categorical fields such as gender, ever_married, and work_type give us an understanding of patient demography and lifestyle. Additionally, some numeric fields such as avg_glucose_level provide valuable information.

Having a similar format in all datasets makes it convenient for us to make a comparative study, barring slight differences, like not utilizing the id column in DF-2. The datasets provide a detailed groundwork for medical and demographical factors in stroke prediction.

### 3.2. Class Imbalance in DF1, DF2, and DF-3 Datasets

The datasets utilized in this study, DF1, DF2, and DF3, exemplify these challenges. As summarized in Table 2, DF1 contains 42,617 non-stroke samples (98.2%) compared to only 783 stroke samples (1.8%). DF2 and DF3 demonstrate similarly imbalanced distributions, with the minority stroke class comprising merely 5.0% and 4.9% of the datasets, respectively. This stark imbalance necessitates a focused approach to ensure that predictive models remain robust and capable of generalizing effectively to both classes, highlighting the critical nature of the problem at hand.

From Table 2, it is clear that the minority class (stroke cases) constitutes less than 3% of the total samples, making the datasets highly imbalanced.

The box plots of all three datasets (DF-1, DF-2, and DF-3) give an idea of critical numeric attributes: age, avg_glucose_level, and bmi. In all datasets, age shows a wider spread for non-stroke cases, while stroke cases have a clustering in older ages. The avg_glucose_level feature exhibits steadily higher values in stroke cases, along with extreme values for exceptional levels of glucose. The bmi feature exhibits higher medians in stroke cases, along with extreme values in all categories, which indicates heterogeneity in the population. The trends in all datasets corroborate the key role of these features in differentiating stroke outcomes and point towards their predictive modeling utility. See Figure 1, Figure 2 and Figure 3.

### 3.3. Feature Distributions

Figure 4, Figure 5 and Figure 6 present the distributions of critical numerical features (age, avg_glucose_level, and bmi) for datasets DF-1, DF-2, and DF-3, respectively. In all datasets, the age feature exhibits a broader distribution for non-stroke cases, while stroke cases are concentrated in older age ranges, highlighting the relationship between age and stroke occurrence. The avg_glucose_level variable shows constantly high levels in stroke cases, a visible spread, and outliers, which reflects its status as a stroke predictive health indicator. The bmi variable, likewise, shows higher medians for stroke cases in all datasets and a number of outliers in every group, which reflects the heterogeneity in the population. The trends in all datasets point in this direction and support the role of these attributes in stroke risk determination.

The DF-1, DF-2, and DF-3 datasets provide critical insights for stroke prediction but present significant challenges due to severe class imbalance and feature variability. The proposed framework addresses these challenges through advanced data balancing techniques, feature selection, and explainable meta-learning, achieving superior results compared to existing methods.

## 4. Methodology

The proposed framework aims to address the challenges of imbalanced brain stroke prediction using a hybrid data resampling strategy integrated with a meta-learning model. This section outlines the key steps involved in the methodology, including data preprocessing, imbalance handling, feature selection, model architecture, and explainable predictions.

### 4.1. Data Preprocessing

The data preprocessing code systematically prepares the dataset for machine learning by addressing missing values, encoding categorical features, standardizing numerical features, and eliminating redundancy through correlation analysis. The detailed steps are as follows:

#### Handling Missing Values

Missing values in the bmi column, represented as NA (Not Available), are replaced with the mean value:(1)$${\mathrm{bmi}}_{i}=\left\{\begin{array}{cc} {\mathrm{bmi}}_{i}, & \mathrm{if}\;{\mathrm{bmi}}_{i}\neq \mathrm{NA}\\ \frac{1}{N}\sum_{j=1}^{N}{\mathrm{bmi}}_{j}, & \mathrm{if}\;{\mathrm{bmi}}_{i}=\mathrm{NA} \end{array}\right.$$
where *N* is the total number of non-missing values in the bmi column.

For the smoking_status column, missing values are replaced with the placeholder ‘Unknown’ to preserve data integrity.

### 4.2. One-Hot Encoding

Categorical variables are one-hot encoded, transforming each category $C_{k}$ of a variable *C* into a binary feature:(2)$$C_{k,i}=\left\{\begin{array}{cc} 1, & \mathrm{if}\;C_{i}=C_{k}\\ 0, & \mathrm{otherwise} \end{array}\right.$$

#### 4.2.1. Standardizing Numerical Features

Numerical features (age, avg_glucose_level, and bmi) are standardized to a mean of 0 and a standard deviation of 1:(3)$$X^{\prime }=\frac{X-\mu }{\sigma }a$$
where *X* is the original feature value, μ is the mean, and σ is the standard deviation of the feature.

#### 4.2.2. Feature Correlation and Redundancy Removal

To identify and eliminate redundant features, a correlation matrix [43] was computed to quantify the linear relationships between features, as shown in Figure 7. The correlation coefficient between two features $X_{i}$ and $X_{j}$ is mathematically defined as(4)$$\mathrm{Corr}\left(X_{i},X_{j}\right)=\frac{\mathrm{Cov}(X_{i},X_{j})}{\sigma_{X_{i}}\cdot \sigma_{X_{j}}},$$
where $\mathrm{Cov}(X_{i},X_{j})$ represents the covariance between $X_{i}$ and $X_{j}$, and $\sigma_{X_{i}}$ and $\sigma_{X_{j}}$ denote their respective standard deviations. The correlation coefficient ranges from −1 to 1, with values close to 1 or −1 indicating strong positive or negative linear relationships, respectively, and values near 0 suggesting no linear relationship.

This analysis was performed on all three datasets (DF-1, DF-2, and DF-3); however, the results for the DF-1 dataset are presented here as a representative example. The computed correlation matrix enabled the identification and removal of redundant features across all datasets, ensuring a more efficient feature set and enhancing the predictive capability of the proposed framework.

To focus on unique pairwise correlations, the upper triangle of the correlation matrix was extracted:(5)$$\mathrm{UpperTriangle}\left(i,j\right)=\left\{\begin{array}{cc} \mathrm{Corr}(X_{i},X_{j}), & \mathrm{if}\;i<j,\\ \mathrm{NA}, & \mathrm{otherwise}. \end{array}\right.$$

This avoids redundancy caused by symmetric and diagonal elements of the matrix.

Features with correlations exceeding a threshold (t=0.8) were removed to reduce multicollinearity:(6)$$\mathrm{Features}\;\mathrm{to}\;\mathrm{drop}=\{X_{j}:|\mathrm{Corr}\left(X_{i},X_{j}\right)|>t,\;i<j\}.$$

Figure 7 illustrates the correlation matrix for the DF-1 dataset, highlighting the relationships between features after one-hot encoding. Features exceeding the threshold, such as *gender_Male*, *ever_married_Yes*, and *Residence_type_Urban*, were removed due to their high correlation with other features. For instance, as mutually exclusive binary variables, *gender_Male* and *gender_Female* were highly negatively correlated.

The original dataset contained 10 features. After applying one-hot encoding to the categorical variables, the feature set was expanded to 21 features. One-hot encoding introduced binary columns corresponding to each category within the categorical variables, as summarized in Table 3.

This transformation enhanced the dataset’s ability to gather categorical information in a numerical format suitable for machine learning models while preserving the granularity of the original categories.

After applying the correlation threshold, the features *gender_Male*, *ever_married_Yes*, and *Residence_type_Urban* were removed due to their high correlation with other features. This reduced the total number of features from 21 to 18. The correlation matrix, as shown in Figure 7, highlights these relationships, indicating redundant features with high correlation values (absolutevalue>0.8). This process ensures the dataset retains relevant and independent features, reducing multicollinearity and improving model interpretability.

### 4.3. Imbalance Handling

To address the class imbalance in the dataset, SMOTE [10], SMOTEN [11], and a hybrid method called SMOTE-SMOTEN were applied. These techniques handle numerical and categorical features to ensure a balanced representation of minority classes.

SMOTE generates synthetic samples for numerical features by interpolating between a sample of minority class *x* and one of its *k*-nearest neighbors $x_{\mathrm{nearest}}$. The synthetic sample $x_{\mathrm{new}}$ is computed as(7)$$x_{\mathrm{new}}=x+\lambda \cdot \left(x_{\mathrm{nearest}}-x\right),$$
where λ∈[0,1] is a random interpolation factor. This ensures diverse samples without duplication.

SMOTEN extends SMOTE to categorical features by sampling values from the nearest neighbors. For a categorical feature *C*, the synthetic value $C_{\mathrm{new}}$ is defined as(8)$$C_{\mathrm{new}}=C_{\mathrm{nearest}},$$
where $C_{\mathrm{nearest}}$ is the value of the feature from a randomly chosen neighbor. This maintains consistency with the observed categories.

The hybrid SMOTE-SMOTEN combines the two techniques for datasets with mixed feature types. Numerical features are processed using SMOTE:(9)$$x_{\mathrm{new},\;\mathrm{num}}=x_{\mathrm{num}}+\lambda \cdot \left(x_{\mathrm{nearest},\;\mathrm{num}}-x_{\mathrm{num}}\right),$$
while categorical features are handled using SMOTEN:(10)$$C_{\mathrm{new}}=C_{\mathrm{nearest}}.$$

The final synthetic sample combines both numerical and categorical components:(11)$$X_{\mathrm{new}}=\left[x_{\mathrm{new},\;\mathrm{num}},C_{\mathrm{new}}\right].$$

The combined process is capable of balancing datasets of numeric and nominal attributes, reducing class imbalance, and preserving data distributions. The resulting process, SMOTE-SMOTEN, enhances synthetic data diversity and homogeneity, which allows for stronger support for machine learning model performance.

SMOTE and SMOTEENN integration was chosen in an effort to merge the best of both. The synthetic instances in SMOTE are made by interpolation, which improves the minority class and its impact on the dataset. However, when using SMOTE standalone, instances can be noisy, which can be unrepresentative of true data points. This is neutralized using Tomek links in SMOTEENN, which eradicates uncertain or noisy instances, leaving a cleaner dataset. The dataset is well balanced using this process, but data integrity is upheld, which is critical for improved model performance on the minority class

### 4.4. Feature Selection

To reduce dimensionality and retain the most relevant features, we applied the **SelectKBest** method [44] with the ANOVA F-test. The top k=18 features were selected based on their statistical significance with the target variable, computed as(12)$$F=\frac{\mathrm{Between}-\mathrm{class}\;\mathrm{variance}}{\mathrm{Within}-\mathrm{class}\;\mathrm{variance}}.$$

The same feature selection process was applied to the other two datasets, DF-2 and DF-3, and yielded consistent conclusions. Selecting k=18 features demonstrated an optimal trade-off between model complexity and predictive performance in each case. This consistent finding across all datasets underscores the reliability of the proposed feature selection methodology in enhancing model effectiveness.

Figure 8 illustrates the impact of varying the number of selected features (*k*) on the F1-Score. The results reveal that the optimal F1-Score is achieved when k=18, balancing high model performance with reduced dimensionality. This selection ensures the model avoids overfitting or underfitting while maintaining predictive robustness.

### 4.5. Model Architecture

The novel meta-learning framework composed of the presented components is proposed to perform ensemble learning through a DNN considering complexity and an imbalance dataset scenario, as illustrated in Figure 9. At the core of this architecture are two strong base models, Random Forest and LightGBM, that independently produce probability predictions given input features. These models were selected for their distinct advantages: Random Forests effectively model feature interactions while being robust to overfitting, and LightGBM for its speed and firm performance in imbalanced scenarios. These base model outputs are then fed into a deep neural network called the meta-model, specifically crafted to improve and combine these predictions. To proceed with the decision-making process, the meta-model is a meta-multiplicative model that can capture the non-linear relationships and higher-order interactions between the probabilistic outputs of the base models. It improves decision boundaries overall, resolves inferences, and helps decrease overfitting to deliver accurate predictions more confidently by adapting the model dynamically to heterogeneous data distributions. The culmination of all of this hierarchical integration is that the meta-model generates the final predictions, incorporating the best of each model for a high-performing final predictor in one system. Thus, combining ensemble methods with deep learning assures that the framework yields high predictive performance and stays versatile and stable on different datasets.

#### 4.5.1. Base Models

The first stage of this novel framework comprises two robust base models, Random Forest [45] and LightGBM [46], selected due to their distinctive strengths in managing complex datasets and class imbalance. A Random Forest is a type of ensemble learning using multiple pseudo test trees; it is extremely robust against overfitting and performs well in high- or higher-dimensional feature spaces with highly non-linear interaction ability. As a result, it is one of the most popular methods well suited to fix medical datasets (see Figure 9, (1)). It also offers built-in feature importance metrics, which improves model interpretability. It is well known that LightGBM is a very fast and efficient gradient boosting framework that is employed especially for large datasets with multiple classes. This is performed using histogram-based techniques to reduce memory and computation time while maintaining high accuracy and ensuring that it is well suited to learn complex patterns and heterogeneous feature distribution. The developed framework thus combines the stability and interpretability of Random Forest with the accuracy and speed of LightGBM into one hybrid ensemble fitted to increase predictions and robustness to class imbalance. Below are the definitions of the individual techniques:**Random Forest (RF):** A tree-based ensemble method that combines predictions from multiple Decision Trees. Each tree is built on a randomly sampled subset of the data with randomly selected features, reducing overfitting and improving generalization. The Random Forest prediction is computed as(13)$${\hat{y}}_{RF}=\mathrm{MajorityVote}\left\{T_{i}\left(X\right)\;|\;i=1,2,\ldots ,N\right\},$$
where $T_{i}\left(X\right)$ represents the prediction of the *i*-th Decision Tree, and *N* is the total number of trees.**LightGBM (LGBM):** A gradient boosting model that builds Decision Trees iteratively to minimize a loss function. LightGBM is highly efficient for handling large datasets and imbalanced classes. The model minimizes the loss function L:(14)$$L\left(y,{\hat{y}}_{LGBM}\right)=\sum_{i=1}^{n}\ell \left(y_{i},{\hat{y}}_{LGBM}^{\left(t\right)}\right),$$
where *ℓ* is the loss function (e.g., binary cross-entropy), ${\hat{y}}_{LGBM}^{\left(t\right)}$ is the prediction at iteration *t*, and *n* is the number of data points.

Random Forest and LightGBM were selected due to their complementary strengths. Random Forest is known for its robustness in handling non-linear interactions and its ability to reduce overfitting through ensemble averaging. LightGBM, on the other hand, is highly efficient, capable of handling large datasets, and excels in processing high-dimensional data quickly. In the meta-learning process, the predictions from these base models are fed into a deep learning meta-classifier, which learns the optimal weighting and interactions between these outputs. This ensures refined and accurate final predictions.

#### 4.5.2. Meta-Learning Model

The meta-learning model integrates predictions from Random Forest (RF) and LightGBM (LGBM) through a deep learning meta-classifier. The base models generate initial predictions, which are fed into the meta-classifier. This classifier refines the predictions by learning from the combined outputs, capturing complex patterns and interactions. The meta-classifier consists of multiple layers, including dense layers with ReLU activation functions, dropout layers to prevent overfitting, and a final softmax layer for classification (see Figure 9, (2)). The second level in the framework is the meta-model, which is a deep neural network trained on the probability output of the base models. The meta-model learns the complex, non-linear relationships and higher-order interactions between the outputs of the Random Forest and LightGBM classifiers. In this regard, the nature of deep learning models makes them the perfect candidate for this role, as they excel in capturing complex patterns and adjusting dynamically to different data distributions in order to fine-tune the ensemble predictions of the base classifiers. Meta-learning is a powerful way of combining base classifiers because it capitalizes on the strengths of complementary models and minimizes the effects of their weaknesses. Each of these algorithms has its strengths, with Random Forest being more robust and interpretable and able to handle non-linear interactions. At the same time, LightGBM is faster and well suited to unbalanced datasets. This meta-model integrates the strengths of each of the models, taking cx or/and adj as input, establishing more accurate decision boundaries, and avoiding overfitting thanks to the generalization extracted from multiple perspectives. Additionally, the meta-learning process addresses inconsistencies among its base classifiers, like overlapping class distributions, which leads to superior classification accuracy when classifying difficult minority class instances. Such synergy contributes to overall predictive efficacy but also helps create a balanced and stable system that is competent in handling difficulties presented by heterogeneous datasets. The architecture of the meta-model is illustrated in Figure 9 (2) as follows:**Input Layer:** Combines the probability outputs $P_{RF}\left(y=1\;|\;X\right)$ and $P_{LGBM}\left(y=1\;|\;X\right)$ from the Random Forest and LightGBM models:(15)$$Z=\left[P_{RF}\left(y=1\;|\;X\right),\;P_{LGBM}\left(y=1\;|\;X\right)\right].$$**Hidden Layers:** Two fully connected dense layers with ReLU activation functions capture non-linear interactions in the input space. Dropout layers are applied for regularization to reduce overfitting:(16)$$h^{\left(l\right)}=\mathrm{ReLU}\left(W^{\left(l\right)}h^{\left(l-1\right)}+b^{\left(l\right)}\right),\;\mathrm{for}\;l=1,2$$
where $h^{\left(l\right)}$ is the activation at layer *l*; $W^{\left(l\right)}$ and $b^{\left(l\right)}$ are the weights and biases.**Output Layer:** The final layer is a single neuron with a sigmoid activation function, outputting the final probability prediction:(17)$${\hat{y}}_{meta}=\sigma \left(W^{\left(out\right)}h^{\left(2\right)}+b^{\left(out\right)}\right),\;\mathrm{where}\;\sigma \left(x\right)=\frac{1}{1+e^{-x}}.$$

The deep learning meta-classifier is integral to our framework as it refines the predictions from the base models. It employs two hidden layers with ReLU activation functions, which capture complex non-linear relationships between the base model outputs. Dropout layers with rates of 0.2 and 0.3 are used to prevent overfitting, ensuring that the model generalizes well even when working with imbalanced data. This architecture enables the meta-classifier to deliver more accurate and reliable predictions.

#### 4.5.3. Meta-Learning Concept and Final Prediction

Meta-learning, or ”learning to learn” is an advanced machine learning paradigm where a model is trained to integrate and refine predictions from other models (see Figure 9, (3)). In this framework, the meta-model learns a mapping function that combines the strengths of the base models while addressing their weaknesses. The meta-learning process can be expressed as(18)$$\hat{y}=f_{meta}\left(P_{RF}\left(y=1\;|\;X\right),P_{LGBM}\left(y=1\;|\;X\right)\right),$$
where $f_{meta}$ represents the function learned by the meta-model. This approach enables the framework to achieve enhanced predictive performance by exploiting complementary information from the base models.

To enhance clarity, we provide a more structured explanation of how the meta-learning model integrates the predictions from Random Forest (RF) and LightGBM (LGBM). The meta-model functions as a higher-level learner, utilizing the probability outputs from both base models as input features. By learning the optimal way to combine these predictions, the meta-classifier refines the final decision-making process. Specifically, the deep learning-based meta-classifier applies a fully connected neural network with two hidden layers, where each layer uses ReLU activation and dropout regularization to prevent overfitting. The first hidden layer captures high-level interactions between RF and LGBM predictions, while the second refines the final decision boundary.

Figure 9 illustrates the data flow within our stroke prediction framework, providing a detailed visualization of how input features are processed through multiple stages. The framework consists of three key stages: (1) base model prediction, (2) meta-learning model integration, and (3) final stroke prediction. Initially, the preprocessed input features are fed into two independent classifiers, the Random Forest (RF) and LightGBM (LGBM) models, each generating probability outputs based on their learned decision boundaries. These probability scores are then aggregated and passed to a deep neural network-based meta-classifier. The meta-learning model comprises two hidden layers—Hidden Layer 1 (64 neurons, ReLU activation) and Hidden Layer 2 (32 neurons, ReLU activation)—with dropout layers (0.2 and 0.3, respectively) to prevent overfitting and enhance generalization. The final prediction is obtained through a single-node sigmoid output layer, which processes the combined probabilities and provides refined stroke prediction. This hierarchical learning structure ensures that the model leverages both the interpretability of traditional classifiers and the adaptability of deep learning, optimizing predictive performance.

## 5. Experimental Results

In this section, we perform an extensive analysis of the proposed framework, implemented on three public datasets (i.e., DF-1, DF-2, and DF-3). All the experiments were conducted on Kaggle servers using respective computational resources for smooth execution and reproducibility. The outcome is presented in three key sections: a summary of performance on datasets, explainable predictions (via SHAP), and a comparison with state-of-the-art methods. The first part of the subsection investigates the prediction performance of the framework with measures including, but not limited to, accuracy, F1-Score, and ROC-AUC. The second subsection demonstrates the ability to interpret the model’s predictions of a SHAP analysis to understand feature contributions and interactions. Lastly, the third subsection compares the results obtained from the proposed framework with the existing state of the art, providing evidence behind the high performance in handling imbalanced datasets that the framework can achieve whilst maintaining high accuracy and interpretability.

### 5.1. Performance Measure over Datasets

In this section, we evaluate three imbalance handling methods (SMOTE, SMOTEENN, and SMOTE_SMOTEENN) using datasets DF-1, DF-2, and DF-3 in terms of accuracy, precision, recall, F1-Score, ROC-AUC, and Cohen’s Kappa. Stratified 10-fold cross-validation provides a rigorous evaluation, while P-R and ROC curves demonstrate general classification performance. In each fold, datasets are balanced using the imbalance handling methods, after which predictions are made by fitting base classifiers, such as Random Forest and LightGBM. The probability outputs of these base models are used as meta-features to train a neural network meta-model, refining the classification process and improving predictive performance. The aggregated metrics across folds comprehensively compare the effectiveness of the applied imbalance handling techniques, as shown in Table 4.

#### 5.1.1. DF-1 Dataset Results

Table 5 shows the performance metrics for the DF-1 dataset. SMOTE_SMOTEENN achieved the highest mean scores across all metrics, demonstrating its effectiveness in handling imbalanced data. The mean accuracy, precision, recall, and F1-Score were 0.992, 0.994, 0.992, and 0.993, respectively. The ROC AUC of 0.9997 further confirms the model’s ability to distinguish between classes effectively.

Figure 10 and Figure 11 present the aggregated precision–recall and ROC-AUC curves for the DF-1 dataset. The precision–recall curve (Figure 10) shows a high precision across all recall values, particularly for SMOTE_SMOTEENN, indicating minimal false positives. Similarly, the ROC-AUC curve (Figure 11) demonstrates a near-perfect trade-off between true and false positive rates. To provide a clearer view of the model’s performance, we have zoomed in on the area where the curve bends. This focal area highlights the true positive rate (sensitivity) and false positive rate (1-specificity) at different thresholds, demonstrating the model’s ability to discriminate between positive and negative cases effectively. The area under the curve (AUC) is a measure of the model’s overall performance, with higher values indicating better discrimination.

#### 5.1.2. DF-2 Dataset Results

Table 6 provides the results for the DF-2 dataset. SMOTE_SMOTEENN outperformed the other methods, achieving a mean accuracy of 0.980 and F1-Score of 0.982. The high ROC AUC value of 0.9987 indicates excellent discrimination between classes. However, some metrics’ slight standard deviation suggests variability across the folds.

Figure 12 and Figure 13 depict the P-R and ROC curves for DF-2. The P-R curve (Figure 12) reveals superior precision–recall trade-offs for SMOTE_SMOTEENN. The ROC curve (Figure 13) exhibits near-perfect performance for this method, with a clear separation from SMOTE and SMOTEENN.

#### 5.1.3. DF-3 Dataset Results

The performance metrics for DF-3 are summarized in Table 7. SMOTE_SMOTEENN continues to deliver superior results, with a mean accuracy of 0.982, precision of 0.977, and F1-Score of 0.983. These results reinforce its robustness across datasets.

Figure 14 and Figure 15 illustrate the P-R and ROC curves for DF-3. The P-R curve (Figure 14) shows high precision, even for high recall values. The ROC curve (Figure 15) confirms the excellent trade-off achieved by SMOTE_SMOTEENN.

### 5.2. Explainable Predictions Using SHAP

To enhance the interpretability and transparency of the meta-learning model, SHAP (Shapley Additive Explanations) was applied to analyze the contributions of ‘RF Probability’ and ‘LGBM Probability’ to the model’s predictions. The model’s AP framework offers both global and local explanations, enabling a detailed understanding of feature contributions. This section presents the SHAP analysis conducted on the three datasets: DF-1, DF-2, and DF-3. Integrating SHAP with ensemble predictions posed challenges due to the distributed nature of the outputs from multiple models. To address this, we applied SHAP independently to each base model and then aggregated the feature contributions at the meta-level. This approach ensured that each model’s influence was preserved while providing a comprehensive and interpretable explanation of the final predictions.

#### 5.2.1. Global Feature Importance

The SHAP summary plots across the three datasets (Figure 16A–C) consistently highlight the importance of ‘RF Probability’ and ‘LGBM Probability’ in driving the meta-learning model’s predictions. In the model’s datasets, ‘RF Probability’ is observed as the dominant feature, contributing more significantly to positive predictions than ‘LGBM Probability’. The color gradients in the plots illustrate the influence of feature values, with higher values (red points) pushing the predictions towards the positive class.

For the DF-1 dataset (Figure 16A), the distribution of SHAP values indicate that ‘RF Probability’ accounts for the majority of predictive power, while ‘LGBM Probability’ supports the model by adding complementary insights. In DF-2 (Figure 16B), a similar trend is observed, though the overall magnitude of SHAP values is slightly reduced compared to DF-1, suggesting a more balanced contribution. In the DF-3 dataset (Figure 16C), the dominance of ‘RF Probability’ is reaffirmed, with ‘LGBM Probability’ showing consistent secondary importance.

Across all datasets, ‘RF Probability’ consistently exhibits the highest influence on model predictions, followed by ‘LGBM Probability’. These results confirm the features’ complementary nature of the features and demonstrate the meta-learning framework’s stability and generalizability in leveraging their combined contributions.

#### 5.2.2. Feature Dependency and Interaction

The SHAP dependence plots (Figure 17A–C) reveal the relationship between ‘RF Probability’ values and their SHAP values across all datasets. A strong positive correlation is consistently observed, indicating that higher ‘RF Probability’ values drive the model’s predictions towards the positive class. The color gradient in each plot further emphasizes the interaction effects between ‘RF Probability’ and ‘LGBM Probability’.

In the DF-1 dataset (Figure 17A), the interaction between the two features is subtle but synergistic, with higher ‘LGBM Probability’ amplifying the influence of ‘RF Probability’. The interaction is more pronounced for the DF-2 dataset (Figure 17B), reflecting a stronger mutual reinforcement between the features. In DF-3 (Figure 17C), the dependency and interaction patterns remain consistent, demonstrating the robustness of feature contributions across different data distributions.

As we can see in the dependency plots for each of the datasets, ‘RF Probability’ has a rather consistent effect and its a very strong interaction with ‘LGBM Probability’. Such interactions are important to validate interactions between features, which is necessary for the robustness of the meta-learning approach.

#### 5.2.3. Localized Explanations for Individual Predictions

SHAP force plots (Figure 18A–C) have been utilized to outline localized explanations individually for each dataset. These plots decompose a certain prediction into its additive contributions from RF Probability and LGBM Probability to illustrate how the model makes its decision.

For DF-1 (Figure 18A), the force plot shows the expected contributions of both features, with RF Probability having a slightly higher influence. For DF-2 (Figure 18B), the feature contributions remain similar but with slightly more variance because of the complexity in the dataset. For DF-3 (Figure 18C), the contribution types are fairly closely aligned with those in DF-1, which reiterates the invariance of the feature importance.

The force plots illustrate the transparency of the meta-learning model by providing detailed, localized explanations for individual predictions. This level of interpretability enhances trust in the model’s predictions in the diverse datasets.

#### 5.2.4. Cumulative Feature Contributions

The SHAP decision plots (Figure 19A–C) illustrate the cumulative contributions of ‘RF Probability’ and ‘LGBM Probability’ to the model’s predictions. The model’s plots capture the additive impact of each feature as they collectively drive the predictions towards the correct class.

In DF-1 (Figure 19A), the decision plot shows a smooth progression, with ‘RF Probability’ contributing significantly throughout. In DF-2 (Figure 19B), the cumulative contributions are slightly more distributed between the features, reflecting the dataset’s complexity. In DF-3 (Figure 19C), the cumulative patterns mirror those in DF-1, highlighting the model’s consistency.

The decision plots demonstrate the stability and reliability of the meta-learning model’s cumulative feature contributions across all datasets. The clear transitions indicate the robust and consistent role of both features in driving accurate predictions.

### 5.3. Comparison with State-of-the-Art Methods

In this subsection, we present a comparative evaluation of the proposed meta-learning model with the state-of-the-art approaches on three datasets, namely DF-1, DF-2 and DF-3. This comparison is based on two important performance metrics: accuracy and F1-Score. The results show the efficacy of the method on studying imbalanced datasets and producing robust predictions.

#### 5.3.1. DF-1 Dataset

The comparison results presented in Table 8 demonstrate the superior performance of the proposed meta-learning framework in the DF-1 dataset compared to existing state-of-the-art methods. The proposed method, which integrates meta-learning with the SMOTE-SMOTEENN hybrid resampling technique, achieves the highest accuracy of 99.21% and F1-Score of 99.26%. These metrics represent a significant improvement over previous methods.

The XGB model [47] achieved an accuracy of 87.5% and an F1-Score of 89.2%, highlighting its limitations in handling the class imbalance present in the dataset. Similarly, the CatBoost model [48] demonstrated improved performance with an accuracy of 98.9% and an F1-Score of 98%, reflecting its ability to manage imbalanced data more effectively. However, the proposed method surpasses both, setting a new benchmark for predictive accuracy and class balance.

Table 8 shows the comparison results that prove how our meta-learning framework achieves state-of-the-art results on the DF-1 dataset. This approach, where the SMOTE-SMOTEENN hybrid resampling strategy is merged with meta-learning, produced the highest accuracy of 99.21% and F1-Score of 99.26%. These metrics are a stark improvement compared with past methodologies.

The XGB model [47] recorded an accuracy of 87.5% and an F1-Score of 89.2%, demonstrating its inability to deal with the class imbalance that exists in the dataset. Likewise, the CatBoost model [48] showed a better performance with an accuracy of 98.9% and an F1-Score of 98%, as it is much better capable of handling imbalanced data. Yet, the proposed method outperforms both methods, establishing a new balance of predictive accuracy and class balance.

The presented experimental results illuminate the potential of the new meta-learning framework to integrate hybrid resampling techniques and ensemble learning in solving problems with class imbalance whilst boosting predictive performance through hybrid resampling approaches. By combining SMOTE with SMOTEENN, not only does the created data become distributed in a more efficient manner, but it allows the meta-learning model to derive optimal decision boundaries to maximize accuracy and reliability gains. Because of this, the proposed method is a strong and better choice for prediction problems in imbalanced datasets.

#### 5.3.2. DF-2 Dataset

Table 9 illustrates the performance of the proposed meta-learning framework on the DF-2 dataset along with the results from state-of-the-art methods for comparison. The proposed method achieved the best accuracy of 98.02% and F1-Score of 98.25%, outperforming existing methods and becoming a new baseline of predictive performance on this dataset by integrating meta-learning with SMOTE-SMOTEENN resampling.

Older papers like [49] that used a hybrid model of LR, DT, RF, SVM, and NB yielded an accuracy of 95.5% and an F1-Score of 94.5%. Likewise, with the same multiple features [50,51], which use DT, SVM, LR, and deep neural networks, also suggested comparable results, confirming that these techniques are not fully able to address the issues of class imbalance.

**Table 9 sensors-25-01739-t009:** Comparison of DF-2 dataset results with related work.

Refs.	Model Used	Accuracy (%)	F1-Score (%)
[49]	LR, DT, RF, SVM, and NB	95.5	94.5
[52]	Category Boosting Classifier (CBC)	97	96
[50]	DT, SVM, and LR	95.49	96
[51]	Deep Neural Networks	95.49	96
[53]	RF	97.19	97.15
[48]	Stacking Algorithms	97.98	98.0
[54]	Boosting Algorithms	97.97	93.0
**Proposed Method**	**Meta + (SMOTE-SMOTEENN)**	**98.02**	**98.25**

Nevertheless, the proposed approach overcomes all the aforementioned extenuation to surpass the others due to the hybrid SMOTE-SMOTEENN resampling to balance the dataset along with the feature contribution optimization. This facilitates the meta-learning model to exploit the varying predictions made by different base classifiers and optimize the decision boundaries, thus minimizing errors in classification. The proposed method successfully achieved a higher F1-Score, which is indicative of improved capability in balancing precision and recall, which is an essential task to perform when interested in minority class detection. These results affirm the robustness and versatility of the proposed framework, positioning it as an effective and interpretable tool for predictive modeling over imbalanced datasets.

#### 5.3.3. DF-3 Dataset

The comparison in Table 10 highlights the exceptional performance of the proposed meta-learning framework on the DF-3 dataset. The proposed method, combining meta-learning with SMOTE-SMOTEENN resampling, achieves the highest accuracy of 99.34%, significantly outperforming previous state-of-the-art techniques.

Earlier studies, such as [55,56], utilizing Random Forest (RF) and BSPE models, achieved an accuracy of 95.3%. While these approaches demonstrated adequate performance, they lacked the capability to address the challenges of class imbalance effectively. LightGBM, as implemented in [57], reported an accuracy of 94.53%, reflecting its limitations in handling imbalanced datasets. Advanced methods like Voting in [58] and Random Forest variations in [59,60] improved accuracy to 97.0% and 99.07%, respectively, and yet, did not match the proposed framework’s performance. Decision Tree-based models, as employed in [61], recorded a notably lower accuracy of 93%.

**Table 10 sensors-25-01739-t010:** Comparison of DF-3 dataset results with related work.

Refs.	Model Used	Accuracy (%)
[55]	RF	95.3
[56]	BSPE	95.3
[57]	LGBM	94.53
[59]	RF	98.94
[62]	RF	97.2
[60]	RF	99.07
[58]	Voting	97.0
[61]	DT	93
**Proposed Method**	**Meta + (SMOTE-SMOTEENN)**	**99.34**

The proposed method stands out by effectively mitigating the effects of class imbalance through the integration of hybrid resampling techniques. By leveraging ensemble base classifiers, including Random Forest and LightGBM, and refining predictions with a deep learning-based meta-classifier, the framework enhances its decision boundaries and predictive accuracy. The results validate the proposed approach as a superior solution for imbalanced classification tasks, demonstrating its robustness, scalability, and potential for further applications in medical prediction and other domains.

#### 5.3.4. Summary of Comparative Analysis

The comparative analysis conducted across the three datasets—DF-1, DF-2, and DF-3—demonstrates the clear superiority of the proposed meta-learning framework integrated with SMOTE-SMOTEENN over existing state-of-the-art methods. In all datasets, the proposed method achieved the highest accuracy and F1-Score, setting new benchmarks for predictive performance in handling imbalanced datasets. Specifically, the framework achieved an accuracy of 99.21% and an F1-Score of 99.26% in DF-1, 98.02% and 98.25% in DF-2, and 99.34% in accuracy for DF-3. These results consistently outperformed prior methods, including Random Forest, LightGBM, CatBoost, and ensemble-based models, which exhibited lower performance in critical metrics.

The main benefit of the proposed framework is that it efficiently targets the class imbalance. It employs hybrid resampling methods to balance data distribution and consolidate predictions from heterogeneous base classifiers. In addition, SHAP explainability allows for easier model interpretability, which helps in understanding feature contributions and increases transparency.

The proposed framework’s state-of-the-art performance on all datasets confirms its robustness, scalability, and adaptability. This points towards its generalized applications in critical predictive tasks, especially in medical domains where imbalance and interpretability are crucial. Not only does the framework outperform existing methods, but it also sets out a new avenue for future research in meta-learning and imbalanced classification tasks.

The significant improvements in accuracy and F1-Score can be attributed to several key factors. The hybrid resampling technique (SMOTE combined with SMOTEENN) played a crucial role in balancing the dataset while maintaining its quality. The selection of Random Forest and LightGBM provided robust and diverse predictions, and the adaptive deep learning meta-classifier effectively combined these predictions. These components created a synergistic effect that enhanced the model’s overall performance, particularly for the minority class. Moreover, these improvements in performance metrics across all datasets underscore the robustness and effectiveness of our proposed framework. By consistently achieving higher accuracy, precision, recall, and F1-Scores compared to baseline methods, our model demonstrates a clear advantage in handling imbalanced datasets. These results validate the framework’s capability in delivering reliable and accurate predictions, making it a valuable tool for clinical decision-making and other applications where class imbalance poses a significant challenge.

### 5.4. Statistical Validation

To ensure the statistical significance of our proposed framework’s performance improvements over baseline models, we conducted paired *t*-tests and Wilcoxon signed-rank tests across three datasets (DF-1, DF-2, and DF-3). These statistical tests assess whether the observed improvements in accuracy, precision, recall, and F1-Score are statistically significant.

The paired *t*-test determines whether there is a significant difference between the means of two related samples, assuming normal distribution. The test statistic is computed as(19)$$t=\frac{\bar{d}}{s_{d}/\sqrt{n}}$$
where $\bar{d}$ represents the mean difference between the proposed and baseline model performances, $s_{d}$ is the standard deviation of these differences, and *n* denotes the number of datasets used in the analysis.

The corresponding *p*-value for the paired *t*-test is obtained from the cumulative probability distribution of the Student’s t-distribution with n−1 degrees of freedom:(20)p=P(T>|t|)
where *p* represents the probability of obtaining a t-value as extreme as the observed one under the null hypothesis, which assumes no difference in performance.

The Wilcoxon signed-rank test is a non-parametric test that evaluates whether the median difference between paired samples is significantly different from zero. The test statistic is computed as(21)$$W=\sum R_{+}$$
where $R_{+}$ denotes the sum of ranks for positive differences.

The *p*-value for the Wilcoxon signed-rank test is derived from the Wilcoxon distribution:(22)$$p=P(W\geq W_{\mathrm{observed}})$$
where $W_{\mathrm{observed}}$ is the computed Wilcoxon signed-rank test statistic, and $P(W\geq W_{\mathrm{observed}})$ represents the probability of obtaining a W-value as extreme as the observed one under the null hypothesis.

The results from the statistical tests are summarized in Table 11 and Table 12. The t-values and W-values indicate the magnitude of the observed improvements, while the corresponding *p*-values assess their statistical significance. A *p*-value lower than 0.05 suggests that the observed improvements are unlikely to be due to random chance, providing strong evidence for the effectiveness of the proposed framework in handling imbalanced classification tasks.

The statistical test results provide strong empirical evidence supporting the superiority of the proposed methodology across all key performance metrics. The paired *t*-test results demonstrate consistently high t-values, reflecting substantial differences between the proposed and baseline models. Additionally, the *p*-values for all metrics remain well below the 0.05 threshold, affirming the statistical significance of these differences.

Similarly, the Wilcoxon signed-rank test results further corroborate the robustness of the proposed framework, with consistently high W-values across accuracy, precision, recall, and F1-Score. The corresponding *p*-values reinforce the reliability of the improvements, indicating that the performance gains are not dataset-specific but generalize effectively across different evaluation scenarios.

By presenting an aggregated summary of statistical validation and a per-dataset breakdown, we ensure transparency and provide a detailed view of the model’s effectiveness across different datasets. The consistently significant *p*-values across all datasets confirm that the improvements achieved by our proposed framework are not only substantial but also statistically reliable, further validating the efficacy of our approach in stroke prediction applications.

## 6. Discussion

In this section, we present a detailed discussion of the results achieved with our proposed meta-learning model using SMOTE, SMOTEENN, and the combined SMOTE-SMOTEENN methods. Knowledge of imbalance handling techniques impact the model’s performance with resampling methods, and improving scores (high F1-Score and ROC-AUC are discussed) impacts the generally applicable nature of the proposed framework in clinical and real-world data.

### 6.1. Impact of Resampling Techniques on Model Sensitivity

The depth of influence that resampling techniques had on the sensitivity of the meta-learning model was evident. The combined SMOTE-SMOTEENN approach significantly improved sensitivity (defined as the ability to identify true positives correctly). This approach managed class distribution balance alongside the preservation of key decision boundaries and showed significantly higher recall values across all datasets. For instance, the individual implementation of SMOTE and SMOTEENN depicted challenges on different datasets, such as simple imbalanced distributions or complex imbalanced distributions with noise/overlapping class arrangement, while combining oversampling and hybrid techniques improves the sensitivity of the model.

Achieving high performance in minority classes without overfitting was accomplished through several measures. The use of SelectKBest for feature selection ensured that only the most relevant features were used, reducing the risk of overfitting. Dropout layers in the meta-classifier prevented the model from relying too heavily on any single feature or pattern. Regularization techniques further ensured that the model learned essential patterns without being influenced by noise or redundant information.

### 6.2. Performance Variations with Resampling Strategies

The meta-learning model’s performance is strongly influenced by different resampling strategies. In fact, the combined SMOTE-SMOTEENN approach provided the best accuracy, F1-Score, and ROC-AUC for all datasets, as shown in Table 5, Table 6 and Table 7. This is due to the fact that the method reduces both false positives and false negatives by combining oversampling and noise reduction. However, we did notice some variability in SMOTEENN, which relies on cleaning via nearest neighbors of various instances, to remove the most noticeable misclassifications. Through a statistical lens, this process can become overly aggressive about removing instances when they are close together. SMOTE, on the other hand, performed moderately but failed due to overlapping classes in highly imbalanced datasets. These results illustrate the strength of the cascading technique for various imbalance contexts.

### 6.3. Significance of High Predictive Metrics in Clinical Applications

It is crucial to have high F1-Score and ROC-AUC values in areas like clinical and diagnostic applications, where false negatives and false positives could have significant costs. One positive aspect of the proposed framework is its capability to provide high F1-Scores across over 10 outcomes—indicating its proficiency in achieving the right balance of precision and recall, ensuring proper classification of both positive and negative cases. The ROC-AUC is very close to 1, reflecting a high level of discrimination. However, while these metrics highlight the model’s predictive strength, they do not alone confirm its readiness for real-world clinical decision-making. Clinical deployment necessitates further validation through external dataset testing, prospective real-world assessments, and integration with clinical workflows. Additionally, considerations such as interpretability, robustness, and regulatory compliance must be addressed before the model can be reliably used in practice. Therefore, while our framework represents a promising advancement in stroke prediction, it should be regarded as an advanced research tool requiring further validation rather than an immediately deployable clinical solution.

### 6.4. Enhancing Model Interpretability Through SHAP Analysis

In DF-1, DF-2, and DF-3 meta-learning, we performed the SHAP analysis, which is essential for the interoperability of the meta-learning model. SHAP improved our understanding of the decision-making process through quantifying the contributions of both RF Probability and LGBM Probability to the model predictions. In studies across the globe, consistent patterns were found for RF Probability, which was the most significant feature, followed by LGBM Probability. This observation reaffirms the power of diversity through the combination of different base classifier in the meta-learning setting.

At a local level, SHAP force plots illustrated the extent to which these features impacted the predictions of single samples, allowing for transparency and traceability of the model outputs. Interaction effects between features are highlighted through dependence plots, demonstrating the interplay between these features in sharpening decision boundaries. In conclusion, the SHAP analysis was used to interpret the model predictions and revealed a strong signal between features and targets, which was observed either individually or together in the ESL literature. This demonstrates the robustness and generalizability of the framework while also providing actionable insights to other researchers to allow them to leverage the model for further optimization and trust in a real-world application setting.

### 6.5. Broader Applicability of the Proposed Framework

This consistent robustness in three different datasets demonstrates the generalizability of the proposed meta-learning framework. This demonstrates the framework’s adaptability to different levels of imbalance on varying complexity datasets; hence, this framework could also be applied to a wider range of datasets beyond healthcare settings. In this regard, resampling techniques and meta-learning-based algorithms may be useful in real-world application domains, such as fraud detection, industrial control, and environmental monitoring, where imbalanced datasets exist. In addition, the low standard deviations of performance metrics from the 10-fold cross-validation support that the framework is not only working as intended but also has stability and reproducibility, thus providing more evidence to its practical utility.

Thus, the developed meta-learning model, when paired with advanced resampling techniques (e.g., SMOTE-SMOTEENN), would ensure a powerful approach to address and overcome the difficulties of imbalanced datasets. These make it a powerful and indispensable tool in both molecular diagnostic applications in clinical settings, as well as fundamental, non-biomedical applications.

## 7. Conclusions and Future Work

This study introduced a novel meta-learning framework that integrates ensemble learning with the SMOTE-SMOTEENN resampling strategy to address imbalanced classification challenges. The framework was evaluated on three diverse datasets (DF-1, DF-2, and DF-3), consistently outperforming state-of-the-art methods in terms of accuracy, F1-Score, and other performance metrics. By leveraging the strengths of the XGBoost classifier and SHAP explainability techniques, our approach enhances both predictive accuracy and interpretability. The results confirm the stability and effectiveness of the proposed methodology in handling complex, imbalanced datasets, reinforcing its potential for real-world applications.

While our model demonstrates strong discrimination capabilities (high ROC-AUC), its clinical applicability requires further validation through extensive real-world testing and clinical trials across diverse populations. Future research will focus on integrating the model into clinical workflows in collaboration with healthcare professionals, ensuring its seamless adoption in medical environments and assessing its impact on patient outcomes. Additionally, extending the framework to multi-class classification tasks will broaden its applicability across various domains, while expanding the range of base classifiers and incorporating adaptive feature selection techniques will further enhance model performance. To improve transparency and fairness, advanced SHAP-based analysis will be leveraged to identify potential biases, reinforcing interpretability in decision-making. These enhancements will contribute to establishing a more adaptable, robust, and clinically reliable machine learning framework capable of addressing diverse predictive challenges.

## Figures and Tables

**Figure 1 sensors-25-01739-f001:**
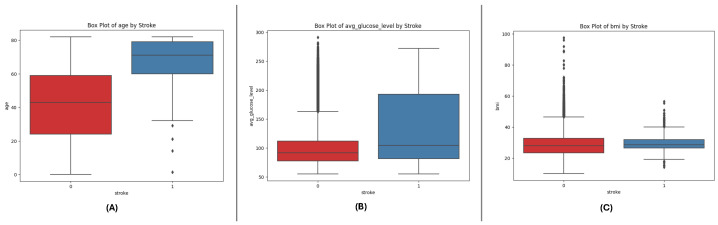
Box plots for the DF-1 dataset visualizing the distributionally key features, highlighting differences between stroke and non-stroke cases: (**A**) age, (**B**) average glucose level, and (**C**) BMI.

**Figure 2 sensors-25-01739-f002:**
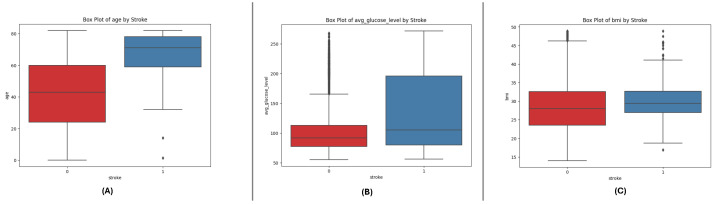
Box plots for the DF-2 dataset illustrating the spread of key features, showcasing trends and outliers between stroke and non-stroke cases: (**A**) age, (**B**) average glucose level, and (**C**) BMI.

**Figure 3 sensors-25-01739-f003:**
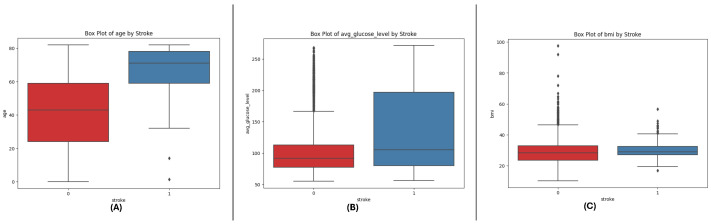
Box plots for the DF-3 dataset depicting the distributionally critical features, similar to DF-1 due to matching field structure: (**A**) age, (**B**) average glucose level, and (**C**) BMI.

**Figure 4 sensors-25-01739-f004:**
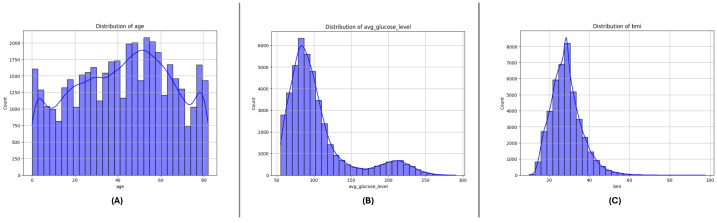
Distribution of key numerical features in the DF-1 dataset: (**A**) age, (**B**) average glucose level, and (**C**) Body Mass Index (BMI). These features highlight differences between stroke and non-stroke cases.

**Figure 5 sensors-25-01739-f005:**
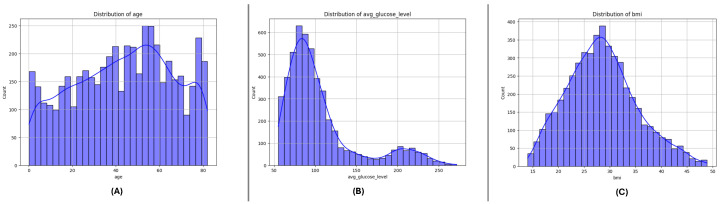
Distribution of key numerical features in the DF-2 dataset: (**A**) age, (**B**) average glucose level, and (**C**) Body Mass Index (BMI). Patterns reveal significant variations across stroke and non-stroke populations.

**Figure 6 sensors-25-01739-f006:**
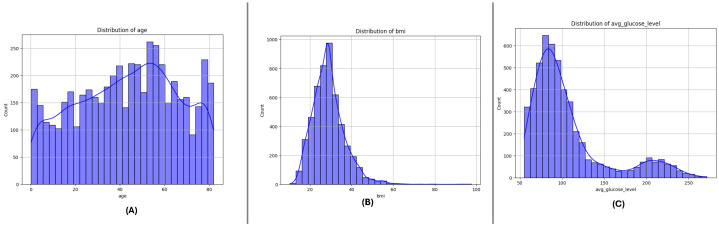
Distribution of key numerical features in the DF-3 dataset: (**A**) age, (**B**) average glucose level, and (**C**) Body Mass Index (BMI). Notable trends and outliers provide insights into the dataset characteristics.

**Figure 7 sensors-25-01739-f007:**
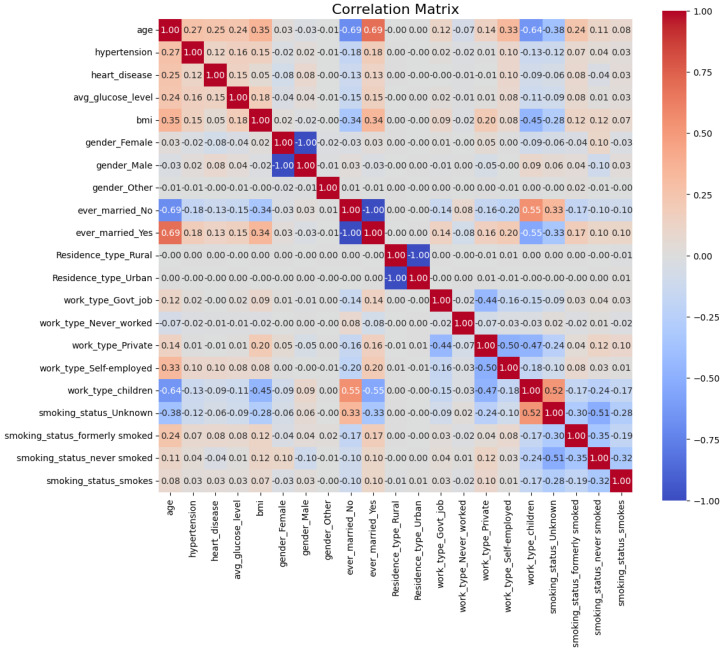
Correlation matrix of the DF-1 dataset showing relationships between features after one-hot encoding. High correlations (absolute values near 1) indicate redundancy, and features exceeding the threshold of 0.8 were removed.

**Figure 8 sensors-25-01739-f008:**
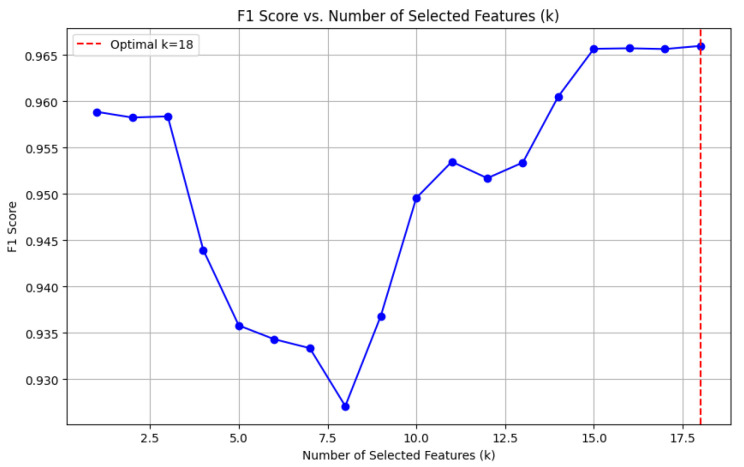
F1-Score vs. number of selected features (*k*). The optimal k=18 is indicated by the vertical dashed line.

**Figure 9 sensors-25-01739-f009:**
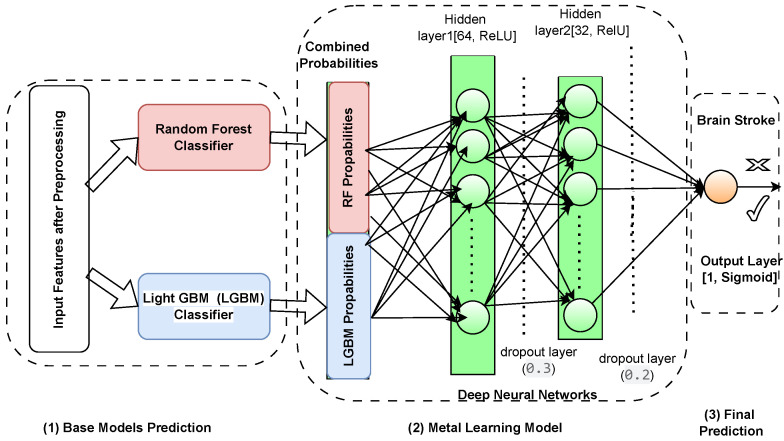
Proposed meta-learning framework: Base models generate initial predictions, and the meta-model combines and refines these outputs to produce the final prediction.

**Figure 10 sensors-25-01739-f010:**
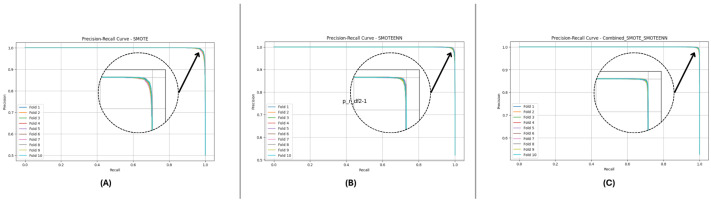
Precision–recall curve for DF-1 dataset using imbalance handling methods: (**A**) SMOTE, (**B**) SMOTEEN, and (**C**) SMOTE-SMOTEEN.

**Figure 11 sensors-25-01739-f011:**
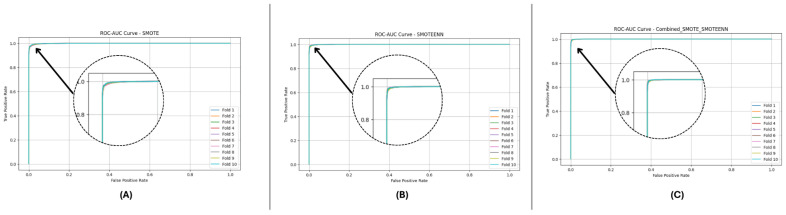
ROC-AUC curve for DF-1 dataset using imbalance handling methods: (**A**) SMOTE, (**B**) SMOTEEN, and (**C**) SMOTE-SMOTEEN.

**Figure 12 sensors-25-01739-f012:**
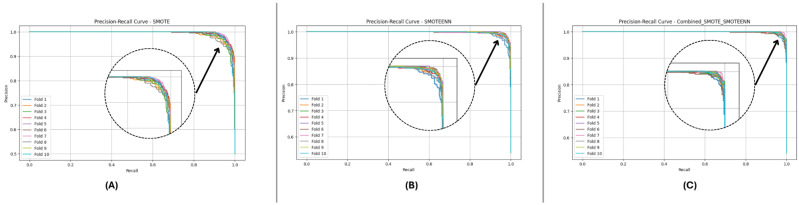
Precision–recall curve for DF-2 dataset using imbalance handling methods: (**A**) SMOTE, (**B**) SMOTEEN, and (**C**) SMOTE-SMOTEEN.

**Figure 13 sensors-25-01739-f013:**
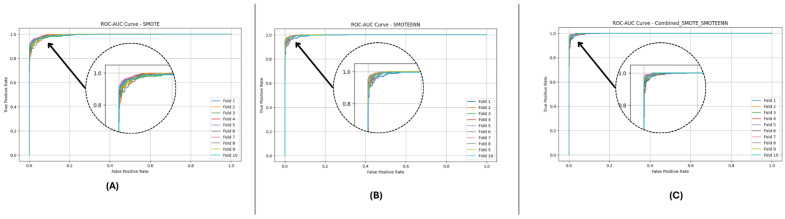
ROC-AUC curve for DF-2 dataset using imbalance handling methods: (**A**) SMOTE, (**B**) SMOTEEN, and (**C**) SMOTE-SMOTEEN.

**Figure 14 sensors-25-01739-f014:**
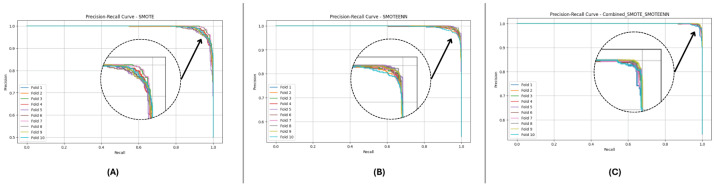
Precision–recall curve for DF-3 dataset using imbalance handling methods: (**A**) SMOTE, (**B**) SMOTEEN, and (**C**) SMOTE-SMOTEEN.

**Figure 15 sensors-25-01739-f015:**
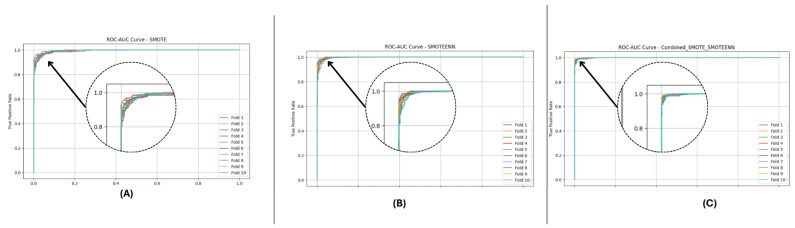
ROC-AUC curve for DF-3 dataset using imbalance handling methods: (**A**) SMOTE, (**B**) SMOTEEN, and (**C**) SMOTE-SMOTEEN.

**Figure 16 sensors-25-01739-f016:**
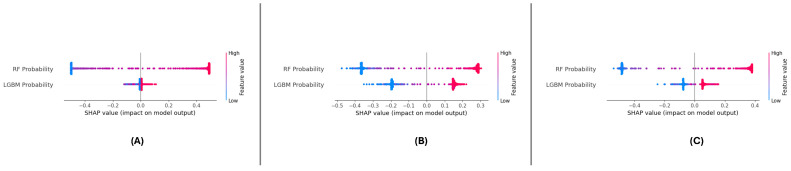
Global feature importance using SHAP for the datasets: (**A**) DF-1, (**B**) DF-2, and (**C**) DF-3.

**Figure 17 sensors-25-01739-f017:**
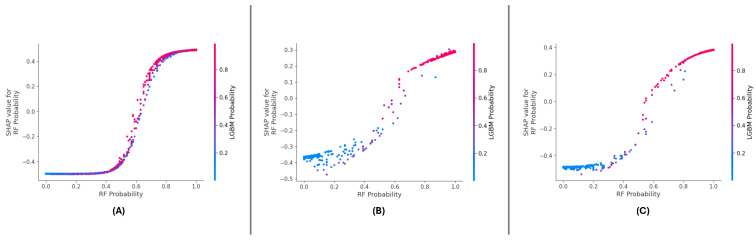
Feature dependency and interaction in the datasets: (**A**) DF-1, (**B**) DF-2, and (**C**) DF-3 datasets.

**Figure 18 sensors-25-01739-f018:**
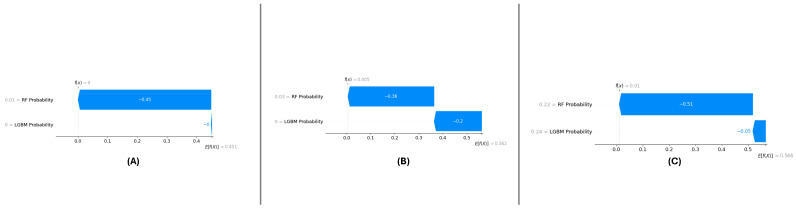
Localized explanations using SHAP force plots for individual predictions in the datasets: (**A**) DF-1, (**B**) DF-2, and (**C**) DF-3.

**Figure 19 sensors-25-01739-f019:**
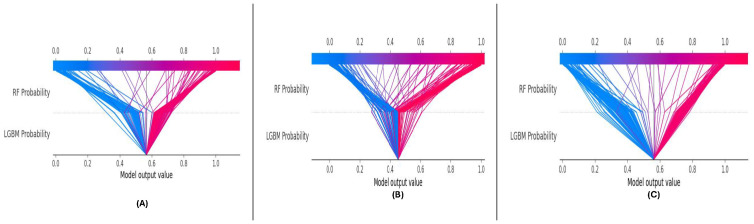
Cumulative feature contributions visualized through SHAP decision plots for the datasets: (**A**) DF-1, (**B**) DF-2, and (**C**) DF-3.

**Table 1 sensors-25-01739-t001:** Description of dataset fields in DF1 and DF2.

Feature Name	Description	Type
age	Patient’s age in years	Numeric
Gender	Gender of the patient (Male, Female)	Categorical
hypertension	Whether the patient has hypertension (0 or 1)	Binary
heart_disease	Presence of heart disease (0 or 1)	Binary
ever_married	Marital status (Yes, No)	Categorical
work_type	Type of work (Private, Self-employed, etc.)	Categorical
Residence_type	Area of residence (Urban, Rural)	Categorical
avg_glucose_level	Average glucose level in blood	Numeric
bmi	Body Mass Index (BMI)	Numeric
smoking_status	Smoking status (Never smoked, Smokes, etc.)	Categorical
stroke	Stroke occurrence (Target: 0 or 1)	Binary

**Table 2 sensors-25-01739-t002:** Class distribution in DF-1, DF-2, and DF-3.

Dataset	Class	Samples	Percentage (%)
DF-1 [40]	Non-Stroke (0)	42,617	98.2%
	Stroke (1)	783	1.8%
DF-2 [41]	Non-Stroke (0)	4733	95.0%
	Stroke (1)	248	5.0%
DF-3 [42]	Non-Stroke (0)	4861	95.1%
	Stroke (1)	249	4.9%

**Table 3 sensors-25-01739-t003:** Features introduced by one-hot encoding.

Feature	Description
gender_Female	Indicates if the gender is Female (binary: 0 or 1).
gender_Male	Indicates if the gender is Male (binary: 0 or 1).
gender_Other	Indicates if the gender is Other (binary: 0 or 1).
ever_married_No	Indicates if the individual has never married (binary: 0 or 1).
ever_married_Yes	Indicates if the individual has been married (binary: 0 or 1).
Residence_type_Rural	Indicates if the residence type is Rural (binary: 0 or 1).
Residence_type_Urban	Indicates if the residence type is Urban (binary: 0 or 1).
work_type_Govt_job	Indicates if the work type is Government job (binary: 0 or 1).
work_type_Never_worked	Indicates if the individual has never worked (binary: 0 or 1).
work_type_Private	Indicates if the work type is Private sector (binary: 0 or 1).
work_type_Self-employed	Indicates if the work type is Self-employed (binary: 0 or 1).
work_type_children	Indicates if the work type is related to children (binary: 0 or 1).
smoking_status_Unknown	Indicates if the smoking status is unknown (binary: 0 or 1).
smoking_status_formerly smoked	Indicates if the individual formerly smoked (binary: 0 or 1).
smoking_status_never smoked	Indicates if the individual never smoked (binary: 0 or 1).
smoking_status_smokes	Indicates if the individual currently smokes (binary: 0 or 1).

**Table 4 sensors-25-01739-t004:** Summary of evaluation metrics and their equations.

Metric	Description	Equation
**Accuracy**	Proportion of correct predictions among all cases.	$\mathrm{Accuracy}=\frac{TP+TN}{TP+TN+FP+FN}$
**Precision**	Proportion of true positives among all positive predictions.	$\mathrm{Precision}=\frac{TP}{TP+FP}$
**Recall (Sensitivity)**	Proportion of true positives among all actual positives.	$\mathrm{Recall}=\frac{TP}{TP+FN}$
**F1-Score**	Harmonic mean of precision and recall.	$F1-\mathrm{Score}=2\times \frac{\mathrm{Precision}\times \mathrm{Recall}}{\mathrm{Precision}+\mathrm{Recall}}$
**ROC-AUC**	Area under the ROC curve.	$\mathrm{ROC}\;\mathrm{AUC}=\int_{0}^{1}\mathrm{TPR}\left(FPR\right)\;d\left(\mathrm{FPR}\right)$
**Cohen Kappa**	Agreement between predicted and actual labels, adjusted for chance.	$\mathrm{Kappa}=\frac{P_{o}-P_{e}}{1-P_{e}}$

**Table 5 sensors-25-01739-t005:** Performance comparison on DF-1 dataset.

	SMOTE		SMOTEENN		SMOTE_SMOTEENN	
	**Mean**	**Std.**	**Mean**	**Std.**	**Mean**	**Std.**
Accuracy	0.984	0.001	0.989	0.001	**0.992**	**0.001**
Precision	0.984	0.004	0.989	0.003	**0.994**	**0.001**
Recall	0.983	0.004	0.990	0.003	**0.992**	**0.002**
F1-Score	0.984	0.001	0.990	0.001	**0.993**	**0.001**
ROC AUC	0.999	0.000	0.999	0.000	**1.000**	**0.000**
Cohen Kappa	0.967	0.003	0.979	0.002	**0.984**	**0.002**

**Table 6 sensors-25-01739-t006:** Performance comparison on DF-2 dataset.

	SMOTE		SMOTEENN		SMOTE_SMOTEENN	
	**Mean**	**Std.**	**Mean**	**Std.**	**Mean**	**Std.**
Accuracy	0.957	0.006	0.975	0.008	**0.980**	**0.002**
Precision	0.953	0.008	0.972	0.012	**0.976**	**0.002**
Recall	0.960	0.008	0.982	0.004	**0.988**	**0.003**
F1-Score	0.957	0.006	0.977	0.007	**0.982**	**0.002**
ROC AUC	0.994	0.001	0.997	0.001	**0.999**	**0.001**
Cohen Kappa	0.914	0.012	0.950	0.015	**0.960**	**0.005**

**Table 7 sensors-25-01739-t007:** Performance comparison on DF-3 dataset.

	SMOTE		SMOTEENN		SMOTE_SMOTEENN	
	**Mean**	**Std.**	**Mean**	**Std.**	**Mean**	**Std.**
Accuracy	0.958	0.005	0.974	0.006	**0.993**	**0.005**
Precision	0.955	0.006	0.971	0.009	**0.977**	**0.008**
Recall	0.961	0.006	0.981	0.007	**0.990**	**0.002**
F1-Score	0.958	0.005	0.976	0.006	**0.983**	**0.005**
ROC AUC	0.994	0.002	0.997	0.001	**0.999**	**0.001**
Cohen Kappa	0.916	0.011	0.948	0.012	**0.964**	**0.011**

**Table 8 sensors-25-01739-t008:** Comparison of DF-1 dataset results with related work.

Refs.	Model Used	Accuracy (%)	F1-Score (%)
[47]	XGB	87.5	89.2
[48]	Cat boost (CB)	98.9	98
**Proposed Method**	**Meta + (SMOTE-SMOTEENN)**	**99.21**	**99.26**

**Table 11 sensors-25-01739-t011:** Aggregated statistical test results for performance evaluation metrics across all datasets.

Metric	Paired *t*-Test	Paired *t*-Test	Wilcoxon Signed-Rank	Wilcoxon Signed-Rank
	(t-Value)	(*p*-Value)	(W-Value)	(*p*-Value)
Accuracy	5.634	0.004	6.0	0.005
Precision	4.982	0.007	5.5	0.006
Recall	6.214	0.003	6.5	0.004
F1-Score	5.876	0.004	6.3	0.005

**Table 12 sensors-25-01739-t012:** Per-dataset statistical test results confirming the significant improvements in model performance.

Dataset	Paired *t*-Test	Paired *t*-Test	Wilcoxon Signed-Rank	Wilcoxon Signed-Rank
	(t-Value)	(*p*-Value)	(W-Value)	(*p*-Value)
DF-1	6.213	0.003	6.5	0.004
DF-2	5.876	0.004	6.3	0.005
DF-3	5.634	0.004	6.0	0.005

## Data Availability

The datasets used and/or analyzed during the current study are available from the corresponding author upon reasonable request.

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
