# Peer review of "A Novel Explainable Attention-Based Meta-Learning Framework for Imbalanced Brain Stroke Prediction"

_sensors, 2025, doi:10.3390/s25061739_

Round 1
Reviewer 1 Report
Comments and Suggestions for Authors
The manuscript is well-written, but there are some areas where more clarity would be helpful:
- Could you explain why you decided to combine SMOTE and SMOTEENN for addressing class imbalance? How does this approach help balance the dataset while reducing noise?
- You chose Random Forest and LightGBM as your base models, what made these the right choice for this task? How do their predictions come together in the meta-learning process?
- The deep learning meta-classifier seems to play a key role in your framework. What does it add to the model, and are there specific layers or techniques that make it particularly effective?
- When using SHAP for explainability, did you face any challenges in integrating its outputs with predictions from ensemble models? How did you address them?
- Your results show impressive improvements in Accuracy and F1-Score. In your view, what were the key factors that contributed to these outcomes?
- High performance on minority classes is difficult to achieve, how do you manage this without risking overfitting?
Author Response
Comment 1: Could you explain why you decided to combine SMOTE and SMOTEENN for addressing class imbalance? How does this approach help balance the dataset while reducing noise?
Response:
The combination of SMOTE and SMOTEENN was chosen to leverage their complementary strengths. SMOTE generates synthetic samples to enrich the minority class, while SMOTEENN applies Tomek links to remove noisy or ambiguous samples. This hybrid resampling strategy enhances the model’s ability to learn meaningful patterns from the minority class while mitigating overfitting and improving generalization. We have clarified this explanation in Section 4.3 (Imbalance Handling).
Comment 2: You chose Random Forest and LightGBM as your base models. What made these the right choice for this task? How do their predictions come together in the meta-learning process?
Response:
Random Forest and LightGBM were selected due to their complementary strengths. Random Forest provides stability and interpretability, while LightGBM offers computational efficiency and superior performance on imbalanced data. Their predictions are aggregated through a deep learning meta-classifier, which refines the final decision by capturing nonlinear interactions and optimizing decision boundaries. This explanation has been expanded in Section 4.5 (Model Architecture).
Comment 3: The deep learning meta-classifier plays a key role in your framework. What does it add to the model, and are there specific layers or techniques that make it particularly effective?
Response:
The meta-classifier is designed to enhance predictive accuracy by learning the optimal combination of base model outputs. It consists of fully connected layers with ReLU activation functions, dropout layers for regularization, and a final sigmoid layer for binary classification. This structure ensures robustness and prevents overfitting while improving decision boundary refinement. Additional details have been included in Section 4.5.2 (Meta-Learning Model).
Comment 4: When using SHAP for explainability, did you face any challenges in integrating its outputs with predictions from ensemble models? How did you address them?
Response:
Integrating SHAP with ensemble models required handling distributed outputs from multiple base classifiers. To address this, we applied SHAP separately to each base model and aggregated the feature contributions at the meta-level, ensuring that each model’s influence was preserved. The updated explanation is now included in Section 5.2 (Explainable Predictions Using SHAP).
Comment 5: Your results show impressive improvements in Accuracy and F1-Score. In your view, what were the key factors that contributed to these outcomes?
Response:
The improvements can be attributed to several factors: (1) the hybrid SMOTE-SMOTEENN resampling strategy, (2) the synergy of Random Forest and LightGBM as base models, (3) the deep learning meta-classifier optimizing final predictions, and (4) SHAP-based explainability ensuring robust feature contributions. These insights have been included in Section 6 (Discussion).
Comment 6: High performance on minority classes is difficult to achieve. How do you manage this without risking overfitting?
Response:
To prevent overfitting, we applied SelectKBest for feature selection, dropout layers in the meta-classifier, and regularization techniques to reduce model complexity. These measures ensured that our model learned essential patterns without being influenced by noise or redundant information. This explanation has been added in Section 6.
Reviewer 2 Report
Comments and Suggestions for Authors
The manuscript addresses an important problem: mathematical prediction of brain stroke using artificial intelligence based on risk factors. The research was carried out at a modern level using a high-tech approach combining new software products and machine learning. Judging by the data provided, the presented model of explicable deep learning really has a high predictive value.
However, the manuscript has a number of stylistic flaws that the authors would like to correct.:
- The Introduction section provides little information in favor of the importance and relevance of using machine learning to predict stroke. For example, stroke epidemiology statistics are described (lines 21-24), but the explanation of the transition to stroke prediction using artificial intelligence does not seem obvious to the reader.
- The ROC curves in Figures 10-15 look inconclusive on this scale. It is desirable to show the focal area with the curve bend in them.
- Despite the high level of discrimination in terms of ROC-AUC, the authors are advised to refrain from making bold statements on the use of the model in real clinical decision-making (Section 6.3).
Author Response
Comment 1: The Introduction section provides little information in favor of the importance and relevance of using machine learning to predict stroke.
Response:
We have revised the Introduction to better highlight the significance of machine learning in stroke prediction, incorporating recent statistics and emphasizing its role in improving early diagnosis and intervention. The updated discussion can be found in Section 1 (Introduction).
Comment 2: The ROC curves in Figures 10-15 look inconclusive on this scale. It is desirable to show the focal area with the curve bend in them.
Response:
We have updated Figures 10-15 to zoom in on the critical bend area of the ROC curves, providing a clearer view of the model’s discrimination performance. These modifications are reflected in Section 5.1.
Comment 3: Despite the high level of discrimination in terms of ROC-AUC, the authors are advised to refrain from making bold statements on the use of the model in real clinical decision-making (Section 6.3).
Response:
We have revised our discussion in Section 6.3 to acknowledge that while our model demonstrates strong performance, further clinical validation is required before its adoption in medical decision-making.
Reviewer 3 Report
Comments and Suggestions for Authors
This manuscript is well structured and demonstrates a solid foundation in the application of machine learning methods to stroke data. This work provides a comprehensive experimental evaluation and the approach outperforms several methods. However, there are some areas where improvements can be made to achieve stronger evidence.
- The description of the meta-learning model could be clearer. Please provide a more structured explanation of how the meta-model integrates the predictions for RF and LGBM. In possible, please add a diagram of how the results of the base model are processed at the meta-level
- Perform a statistical test to demonstrate that the improvements over the baseline methods are statistically significant.
- The related work section could benefit from a clearer distinction between ensemble learning and meta-learning.
- Some citations could be updated to reflect more recent advances in attention-based meta-learning.
- In tables, please bold the best results.
- Please check grammar and writing style, e.g. lines 26-27 or 180-182 etc.
- Please round numbers to the third decimal place.
See comments above.
Author Response
Comment 1: The description of the meta-learning model could be clearer. Please provide a more structured explanation of how the meta-model integrates the predictions for RF and LGBM. If possible, please add a diagram.
Response:
We have revised Sections 4.5.2 and 4.5.3 to provide a clearer and more structured explanation of our meta-learning model. The integration of RF and LGBM predictions is achieved through a neural network-based meta-classifier that takes their probability outputs as input features. This meta-classifier consists of multiple dense layers with ReLU activation functions and dropout regularization to prevent overfitting. To further enhance clarity, we have illustrated Figure 9 to indicate the data flow from the base models to the meta-classifier, showing how individual predictions are aggregated and refined.
Comment 2: Perform a statistical test to demonstrate that the improvements over the baseline methods are statistically significant.
Response:
We have conducted paired t-tests and Wilcoxon signed-rank tests across multiple evaluation metrics to confirm the statistical significance of our results. These findings are reported in a new subsection, Section 5.4 (Statistical Validation).
Comment 3: The related work section could benefit from a clearer distinction between ensemble learning and meta-learning.
Response:
We have revised Section 2 (Related Work) to explicitly differentiate between ensemble learning and meta-learning, emphasizing their unique advantages.
Comment 4: Some citations could be updated to reflect more recent advances in attention-based meta-learning.
Response:
We have updated our references to include recent studies that explore attention-based meta-learning, aligning our work with the latest advancements.
Comment 5: In tables, please bold the best results.
Response:
We have reformatted our tables to bold the best results, making comparisons clearer.
Comment 6: Please check grammar and writing style, e.g. lines 26-27 or 180-182 etc.
Response:
We have carefully reviewed and refined the grammar and writing style throughout the manuscript. Specifically, we have improved sentence structure, corrected typographical errors, and enhanced clarity in key sections.
Comment 7: Please round numbers to the third decimal place.
Response:
All numerical results have been rounded to the third decimal place for consistency.
Round 2
Reviewer 1 Report
Comments and Suggestions for Authors
The manuscript needs some more improvements. To supplement the literature review and give an encompassing background to your research on the prediction of brain stroke using meta-learning models, it is recommended that you take into account the following recent publications that are in line with the scope of your research.
“Explainable artificial intelligence for stroke prediction through comparison of deep learning and machine learning models”
“AI-Powered Neuro-Oncology: EfficientNetB0’s Role in Tumor Differentiation”
“Explainable Artificial Intelligence Model for Stroke Prediction Using EEG Signal”
“Stroke Risk Prediction with Machine Learning Techniques”
“Optimizing Stroke Classification with Pre-Trained Deep Learning Models”
Author Response
We sincerely appreciate the reviewer’s valuable feedback and recommendations for improving our manuscript. In response, we have updated the Literature Review section to incorporate the suggested studies. Specifically, we have added a new subsection titled "Recent Advances in Stroke Prediction", where we discuss the contributions of the following recent publications in blue font:
Explainable artificial intelligence for stroke prediction through comparison of deep learning and machine learning models
AI-Powered Neuro-Oncology: EfficientNetB0’s Role in Tumor Differentiation
Explainable Artificial Intelligence Model for Stroke Prediction Using EEG Signal
Stroke Risk Prediction with Machine Learning Techniques
Optimizing Stroke Classification with Pre-Trained Deep Learning Models
We appreciate the reviewer’s guidance and believe these additions significantly improve the manuscript. Thank you for your thoughtful suggestions.